# Evidencing Fluid Migration of the Crust during the Seismic Swarm by Using 1D Magnetotelluric Monitoring

Carlos A. Vargas [1], Alexander Caneva [1,*], Juan M. Solano [1], Adriana M. Gulisano [2,3,4] and Jaime Villalobos [5]

[1] Department of Geosciences, Universidad Nacional de Colombia, Bogotá 111321, Colombia
[2] Instituto Antártico Argentino, Dirección Nacional del Antártico, Buenos Aires B1650HMK, Argentina
[3] Instituto de Astronomía y Física del Espacio, CONICET, Universidad de Buenos Aires, Buenos Aires C1053ABH, Argentina
[4] Departamento de Física, Facultad Ciencias Exactas y Naturales, Universidad de Buenos Aires, Buenos Aires C1053ABH, Argentina
[5] Department of Physics, Universidad Nacional de Colombia, Bogotá 111321, Colombia
* Correspondence: acanevar@unal.edu.co

**Abstract:** We applied multi–temporal 1D magnetotelluric (MT) surveys to identify space–time anomalies of apparent resistivity ($\rho_a$) in the upper lithosphere in the Antarctic Peninsula (the border between the Antarctic and the Shetland plates). We used time series over several weeks of the natural Earth's electric and magnetic fields registered at one MT station of the Universidad Nacional de Colombia (RSUNAL) located at Seymour–Marambio Island, Antarctica. We associated resistivity anomalies with contrasting earthquake activity. Anomalies of $\rho_a$ were detected almost simultaneously with the beginning of a seismic crisis in the Bransfield Strait, south of King George Island (approximately 85.000 events were reported close to the Orca submarine volcano, with focal depths < 20 km and $M_{WW}$ < 6.9). We explained the origin of these anomalies in response to fluid migration near the place of the fractures linked with the seismic swarm, which could promote disturbances of the pore pressure field that reached some hundreds of km away.

**Keywords:** apparent resistivity; earthquakes; magnetotellurics; electromagnetic anomalies; Antarctic Peninsula; Seymour–Marambio Island; Orca submarine volcano





## 1. Introduction

Physical property anomalies of the lithosphere, for instance, electrical resistivity, were hypothetically associated with earthquake activity in recent decades. In addition, it has been suggested that there are relationships between the magnitude of the earthquakes and the amplitude duration of apparent resistivity ($\rho_a$) anomalies [1,2]. In other cases, several electromagnetic (EM) anomalies were also detected in the ionosphere and subsurface and linked to large–magnitude seismic events [3,4].

Intense seismic activity has been related to changes in pore pressure [5]. These changes possibly are due to the fluid flux as a consequence of variations in the stress field. Hence, the $\rho_a$ changes are expected to reflect temporal pore pressure changes.

Deep mapping of $\rho_a$ is reached by using MT deployments. This method uses natural electrical and magnetic signals recorded at the Earth's surface, allowing us to estimate vertical electrical resistivity profiles in the subsoil by using relationships between orthogonal components of the electric and magnetic fields [6]. MT is not frequently used for monitoring due to the time consumption of data acquisition. However, it appears to be a reliable method for reaching depth resolutions similar to those associated with the earthquake sources in the brittle lithosphere.

However, the Bransfield Strait–Antarctic Peninsula is a region of active tectonics in response to a subduction process of the Shetland Plate under the Antarctic Plate [7]

(Figure 1). The Bransfield Strait defines the geometry of the Bransfield Basin, a back–arc rift basin limited by the Shackleton Fracture Zone and the Hero Fracture Zone located north and south of the Shetland Plate, respectively. Swarms of seismic events near submarine volcanoes suggest current magmatic activity. In addition, normal faulting inferred from focal mechanisms in the strait suggests a transtensional process confirmed by different active rifts.

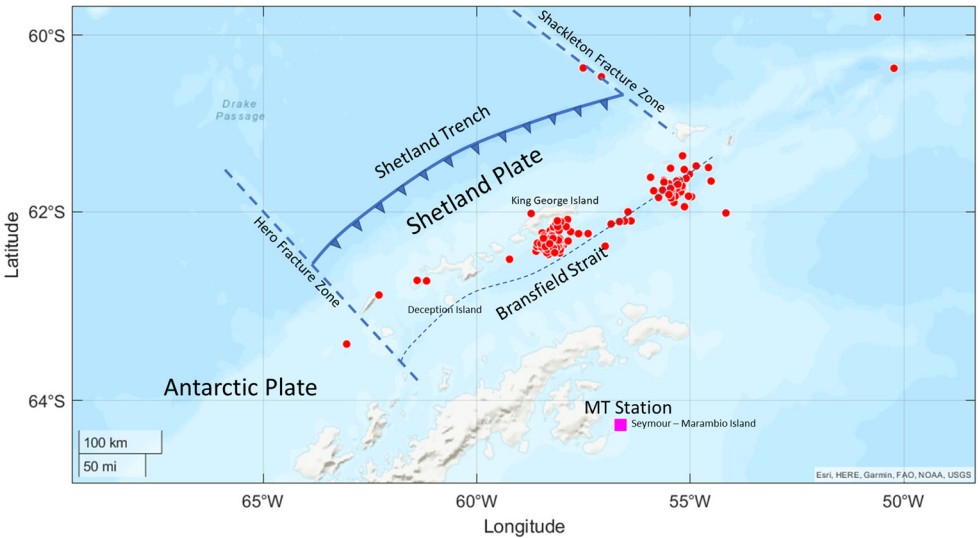

**Figure 1.** The main tectonic features are in the influence area of the Shetland and Antarctic plates. Red points correspond to earthquakes reported by the USGS with $m_b > 4.0$, which have epicentral distances between 200 and 300 km from the MT station installed in the Seymour–Marambio Island (purple square). The axis of the Bransfield Basin is suggested by a thin–dashed line, following the trend of the Bransfield Strait.

Taking advantage of the EM signals generated naturally in this region with low anthropogenic interference, in this work, we estimated space–time variations of the $\rho_a$ in the crust to infer possible fluid migration and the changes in the pore pressure linked to the tectonic source that generated a seismic swarm in the Bransfield Strait, south of King George Island–25 de Mayo Island (Figure 1). According to the German Research Centre for Geosciences Potsdam [8], approximately 85.000 events were detected between August and November 2020 under the Orca submarine volcano, with focal depths < 20 km and a maximum of $m_b = 6.9$.

## 2. Materials and Methods

### 2.1. Magnetotelluric Station in the Seymour–Marambio Island

The $\rho_a$ variations in time and space (depth) in the upper lithosphere were estimated using a permanent MT station located north of Seymour–Marambio Island, Antarctica, near the Multidisciplinary Antarctic Laboratory of the Marambio Island (LAMBI) (Figure 2a,b). It contains sensors to measure the magnetic and electric fields (Figure 2c), such as a Bartington Mag648L triaxial magnetometer with low noise, a range of $\pm60$ μT (resolution of 0.012 nT/count), and four copper ground electrodes of 70 cm each, all located approximately at the same level on a plateau 200 m high. The magnetometer was buried inside the permafrost in a hole dug 75 cm deep to avoid drastic variations in surface temperature. In comparison, the electrodes were percussion buried 80 cm deep with their top connection protected by silicone to prevent corrosion. Each electrode was connected to a 1.2 cm diameter copper cable protected by a polymer resistant to extreme temperatures. The arrangement of the four electrodes made up the two almost orthogonal dipoles NNE (124 m) and EEN (80 m), which maintained the same direction as the magnetometer components. In addition, light

gases such as $CH_4$ and $CO_2$ were measured to evaluate possible new evidence of their massive escape during the seismic swarm.

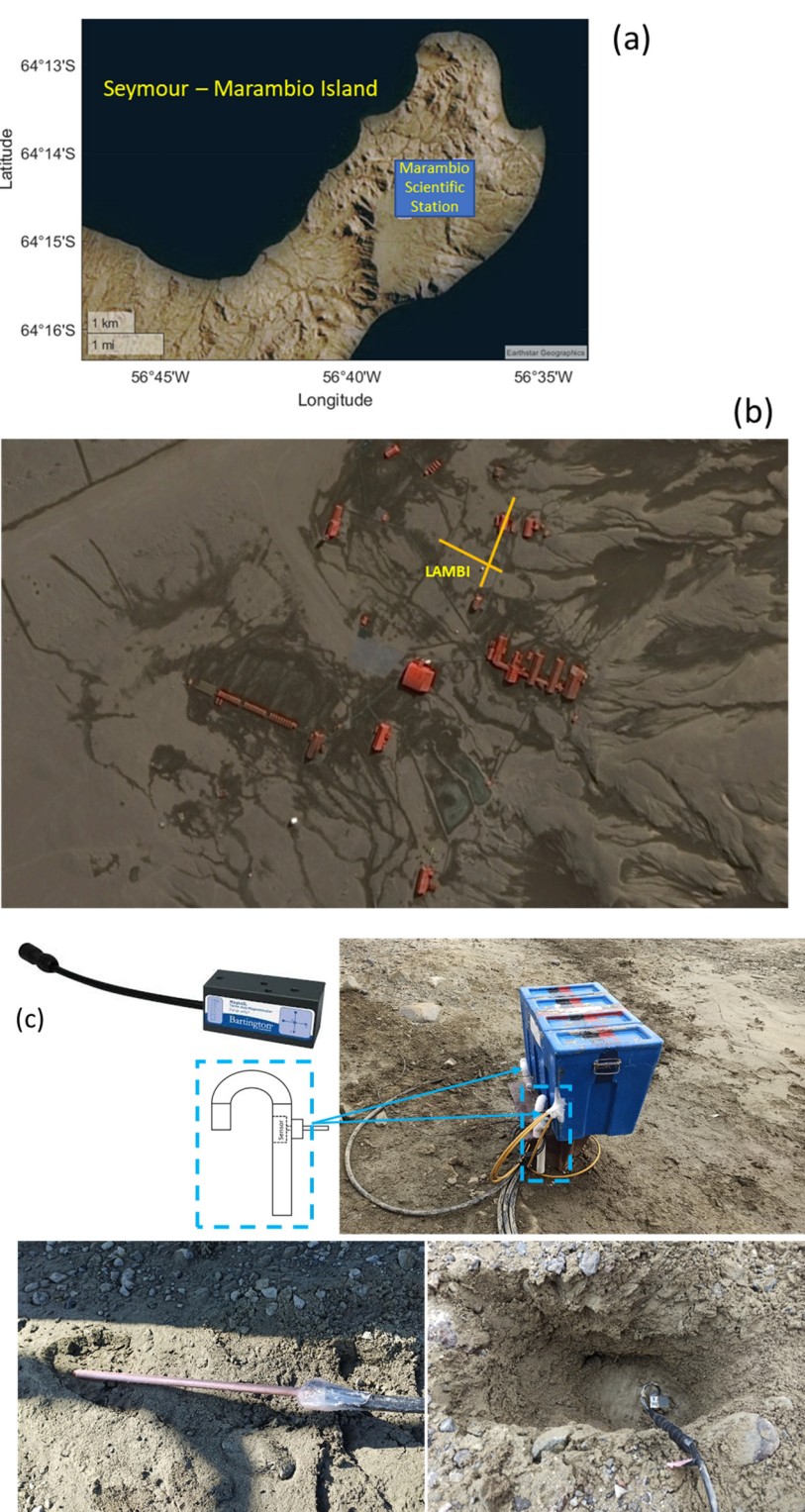

**Figure 2.** (**a**) MT deployment at the north Seymour–Marambio Island. (**b**) NNE (124 m) and EEN (80 m) are the two orthogonal dipoles' locations. (**c**) Picture of the triaxial magnetometer Bartington Mag648L and the snorkel tube where $CH_4$ and $CO_2$ are sensed (upper left). The blue box contains the coupling and digital recording system (upper right). The four dipole electrodes were percussion buried 80 cm deep (lower images). The magnetometer and electrodes were buried in the permafrost.

*2.2. Magnetotelluric Sounding (MT)*

The MT method studied the penetration and propagation of electromagnetic waves inside the Earth, associated with the action of electrical storms and/or the incidence of the solar wind on the Earth [6]. The method was based on measurements taken on the Earth's surface of the natural electric field (employing two perpendicular electric dipoles, Ex and Ey) and the magnetic field (in our case, using a triaxial flux magnetometer with components Bx, By, and Bz).

The Earth's surface partially reflected the fluctuating electromagnetic fields that originated in the ionosphere, and the ionosphere again reflected the returning fields due to their conductive characteristics. It repeatedly happened so that the fields eventually had a strong vertical component and could be considered vertically propagating plane waves, characterized by covering a broad spectrum of frequencies. These fields penetrated the ground and induced telluric (electric) currents, generating secondary magnetic fields. The telluric currents, detected by two pairs of electrodes, each pair of which composed a dipole, were perpendicularly oriented. The three components of the magnetic fields were measured: the vertical component and two horizontal components, parallel to each one of the electrical components [9,10].

This method provided information about the resistivity (conductivity) values for much greater depths than artificial source induction methods. Using long–period signals ranging from tens to thousands of seconds, the MT method reached investigation depths that may sample the entire lithosphere [9]. In this work, the permanent station guaranteed continuous datasets for years and, consequently, large depths of investigation.

Details for estimating the apparent resistivity structure of the subsoils for an instant and using the MT method can be found in [10] and [6]. Because an electromagnetic wave in the subsoil decays its amplitude due to the resistivity of the medium (assuming a homogeneous half–space), the depth at which the amplitude reaches the factor $1/e \approx 0.37$ (skin depth, $\delta$) can be estimated using the expression $\delta(\omega) = \sqrt{2/\omega\mu\sigma}$ [6,10], which can be simplified to:

$$\delta \approx 503\sqrt{\rho T}, \tag{1}$$

where $\omega$ is the angular frequency under the assumption of $e^{i\omega t}$ time dependence, $\mu$ is the magnetic permeability, $\sigma$ is the electrical conductivity, $\rho$ is the resistivity, and $T$ is the period evaluated. Thus, the procedure implemented with the data acquired for this work is briefly described as follows:

(1) Time windows of 5000 s were chosen. An overlap of 500 s was used to estimate the resistivity's temporal evolution. Each time window was tapered with a Hanning window, and its frequency spectrum was calculated using the FFT.

(2) The orthogonal components of the natural electric (**E**) and magnetic (**B**) fields were related to the impedance tensor **Z** of the subsoil in the following way [10]:

$$\begin{pmatrix} E_x \\ E_y \end{pmatrix} = \begin{pmatrix} Z_{xx} & Z_{xy} \\ Z_{yx} & Z_{yy} \end{pmatrix} \begin{pmatrix} \frac{B_x}{\mu_0} \\ \frac{B_y}{\mu_0} \end{pmatrix}, \tag{2}$$

where $\mathbf{E} = \mathbf{Z}\frac{\mathbf{B}}{\mu_0}$ and $\mu_0$ is the magnetic permeability of free space $\left[ \text{V s A}^{-1}\,\text{m}^{-1} = \text{H m}^{-1} \right]$. From the complex coefficients of the frequency spectrum, the $Z_{xy}$ component under the station was estimated.

(3) The Schmucker–Weidelt ($C$) transfer function was estimated as follows:

$$C(\omega) = \frac{E_x}{i\omega B_y} = \frac{Z_{xy}}{i\omega}. \tag{3}$$

(4) The $\rho_a$ was calculated using the following expression:

$$\rho_a(\omega) = |C(\omega)|^2 \mu_0 \omega. \tag{4}$$

(5)　The $\rho_a$ matrix was graphically represented for a given moment and different periods (depths according to Equation (1)), thus guaranteeing a space–time representation of the resistivity field that allowed detecting anomalies concerning a line base of observations.

### 2.3. $K_p$ Index

According to NOAA's Space Weather Prediction Center, this index quantifies the level of impact on the Earth's horizontal magnetic field measured on the surface due to geomagnetic activity, that is, the emission of charges and high solar radiation. It means a significant noise source for our time series, and its activity is classified by the NOAA as follows:

(1)　$K_p < 4$ (weak solar activity);
(2)　$K_p = 4$ (mean solar activity);
(3)　$K_p > 4$ (high solar activity).

An index $K_p > 4$ (high solar activity) would imply a high disturbance in the Earth's magnetic field and a low reliability of the data taken in that time interval. The details of its estimation are presented in the literature on geomagnetism and solar–terrestrial physics [11]. Hence, in addition to the space–time $\rho_a$ representation, we included the evolution of this index.

## 3. Results

Figure 3 synthesizes the mapping of the vertical structure (1D) of the $\rho_a$ along the time, including the $\rho_a$ anomalies ($\Delta\rho_a / \overline{\rho_a}$ %), the $K_p$ index, the depth of the hypocentral solutions of earthquakes with $m_b > 4.0$, and measurements of the emissions of $CH_4$ and $CO_2$ in the Marambio Station. The first significant event of the seismic swarm in the Bransfield Strait, south of King George Island, occurred on 2020/8/29, 12:47:3.768 UTC, Lat 62.4437S, Lon 58.1777W, H = 10 km, and $M_{ww} = 4.9$ (moment W–phase [12]). In terms of real values of $\rho_a$ or anomalies, we detected changes of $\rho_a$ approximately four hours previous to the event. Thirty–one hours later, another event of $M_{ww} = 5.4$ occurred, approximately 25 km from the last event, preceded by a large $\rho_a$ anomaly, which reached almost the surface (Figure 4). Other minor $\rho_a$ anomalies were also detected in this time window with connection to the deeper 1D–$\rho_a$ structure or related to the solar activity as suggested by the $K_p$ indexes. The larger $\rho_a$ anomalies (A and B) did not match with large $K_p$ indexes, meaning that the anomalies were not influenced by solar activity. High values of $\rho_a$ near the surface (upper panel in Figure 3) could be interpreted as the not–well–consolidated permafrost and ice, and the permanent low values of the $\rho_a$ anomalies around 10 km depth could be related to a lithological transition in the local structure of the upper lithosphere near the measurement instrument, which may connect other deeper and shallower fractures that facilitate the fluid migration.

Figure 4 shows details after starting the large anomaly (A) appreciated in Figure 3 on 2020/8/30, where it is possible to observe a shallowing of the $\rho_a$ anomalies that match the beginning of more significant emissions of $CH_4$ and $CO_2$ in the Marambio Station. It suggests that pore pressure perturbations generated by the tectonic stress field disturbances in the region, meaning those that triggered the earthquakes, e.g., events of 2020/8/29, 12:47:3.768 UTC, and 2020/8/30, 19:42:27.389 UTC, with magnitudes $M_{ww} = 4.9$ and $M_{ww} = 5.4$, respectively, could be responsible for the migration of fluids and gases that reached the surface, which may be related to changes in the $\rho_a$ (from deeper sources up to the surface, as suggested by the yellow arrows). Gasses detected by the Marambio Station are typically stored in the permafrost near the measurement instrument [13]. Thus, pore pressure perturbations from the seismic source in the Bransfield Basin could promote their trigger emission in Marambio Island. We also observed that there was a sudden increase in $CO_2$ gas. However, the variation of the $CH_4$ did not have the same trend. We speculate that contrasting concentrations of these gasses near the measurement instrument may explain this behavior.

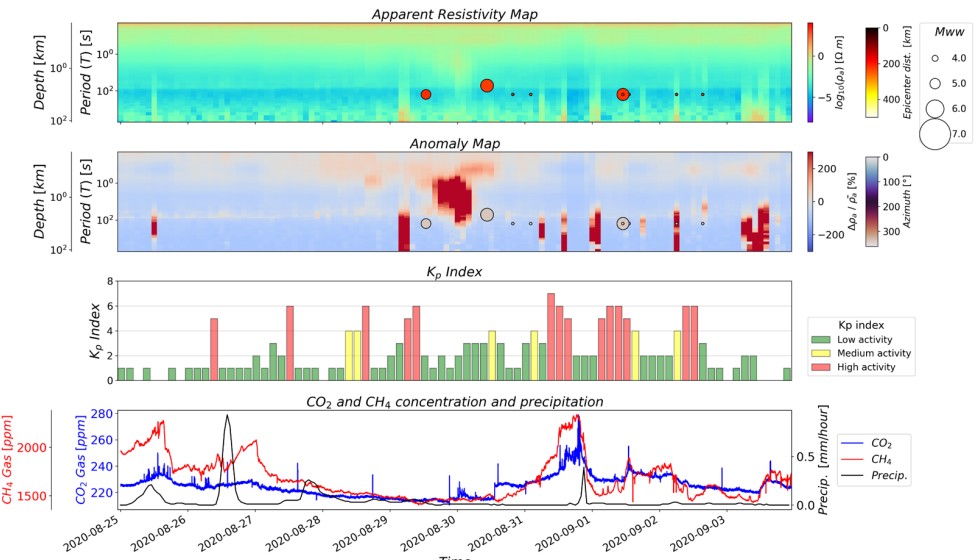

**Figure 3.** Evolution of the 1D–$\rho_a$ structure. The upper panel includes the depth of the hypocentral solutions of earthquakes with $m_b > 4.0$ located in the Bransfield Strait, south of King George Island. The persistent low values of $\rho_a$ in shallow depths may correspond to not–well–consolidated permafrost and/or surficial ice cap (upper panel). The second panel shows anomalies of $\rho_a$. Earthquakes reported by the USGS are in the same seismic swarm (epicentral distances and azimuth angles are almost similar). The third panel shows the $K_p$ index, suggesting that larger $\rho_a$ anomalies (A and N) may not be linked to solar activity. The lower panel shows temporal variations of $CO_2$ and $CH_4$ gas emissions. The extremely low precipitation at the Marambio Station does not seem to affect gas concentrations. Precipitation data are taken from https://power.larc.nasa.gov/data--access--viewer/ (accessed on 15 December 2022).

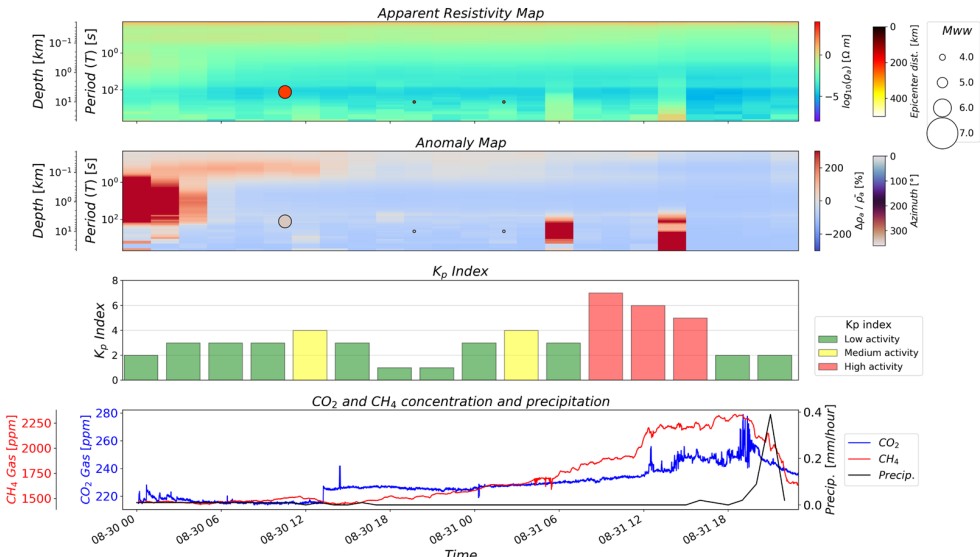

**Figure 4.** Time window after starting large anomaly (A) and presented in Figure 3, which shows a shallowing of the $\rho_a$ anomalies (suggested by yellow arrows in the second panel) that coincide with the beginning of more significant emissions of $CH_4$ and $CO_2$ in the Marambio Station. These gasses are typically stored in the permafrost near the measurement instrument [13], but regional pore pressure perturbations promote their trigger emission. Deeper and strong anomalies in the 1D–$\rho_a$ structure seem to be influenced by solar activity, as shown the $K_p$ index distribution. Precipitation data are taken from https://power.larc.nasa.gov/data--access--viewer/ (accessed on 15 December 2022).

## 4. Discussion

The current geological setting and the tectonic evolution of the Antarctic Peninsula have been addressed during the last decades with important advances. However, heated debates have arisen about the lithospheric structure and geodynamics that underlie this region, the tectono–magmatic activity surrounding it, and the implications of lithosphere–atmosphere interaction [7,13–15]. Figure 5 illustrates the distribution of earthquakes (M > 4.0, reported by the USGS) in the study zone. Focal mechanisms and other hypocentral solutions with depths <50 km are presented on a gravity anomaly map [16], whose resolution only defines the geometry of the Bransfield Basin. We also overlaid the faults interpreted with swath bathymetry data [17] to infer the faulting responsible for the seismic swarm. Because there is no bathymetric evidence that these earthquakes could reach the seabed, the focal mechanisms are the unique tool that informs about an extensional regime with a strike–slip component. This tectonic regime has previously controlled several volcanic emplacements in the Bransfield Strait (e.g., Deception Island and Orca submarine volcano) [14,15]. More recently, the intrusion of 0.26–0.56 km$^3$ of magma has been suggested for the seismic swarm analyzed in this work [8], which could be linked with dramatic fluid migration in the crust and the sea. This figure's heat flux distribution (lower map) [18] shows the epicentral location of the seismic swarm, just in the zone of thermal contrast and throughout the Bransfield basin. As in other regions of the world bordered by several plates, the lithosphere system has dramatic lateral changes in thickness and thermal response [19–21]. In this case, the thermal and thickness lithospheric structures could be related to intense fracturing, forming an efficient porosity system based on microfractures where fluids migrate significant distances or facilitate the strong pore pressure gradients.

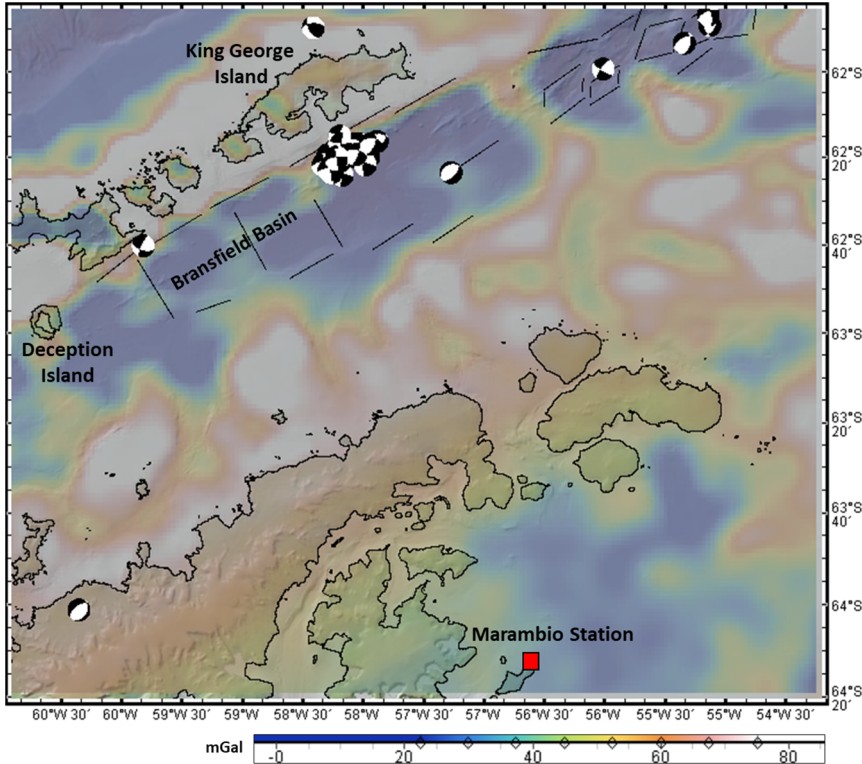

**Figure 5.** *Cont.*

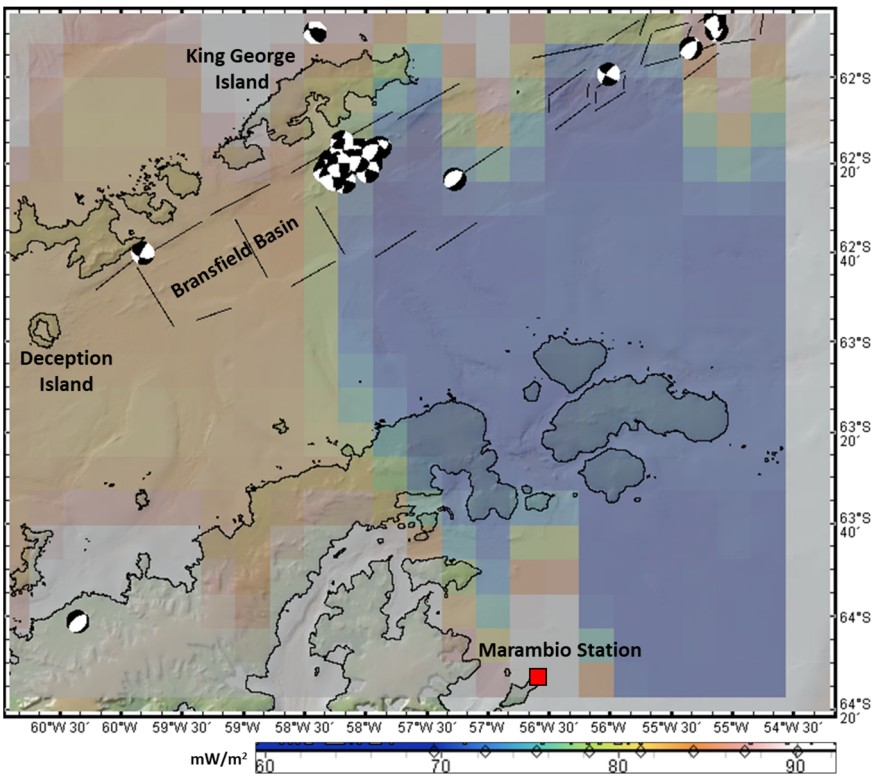

**Figure 5.** Antarctic Peninsula Map, including the archipelagos of the Shetland Plate (north of the Bransfield Basin). The beach balls represent focal mechanisms, and the green circles are some earthquakes with depths <50 km reported by the USGS. The free air anomalies [16] (upper map) define the geometry of the Bransfield Basin. The heat flow distribution [17] (lower map) in the area is <90 mW/m². Suggested faults inside the basin were interpreted with swath bathymetry data [18]. The Marambio station (red square) represents the location of the MT monitoring. Figure made with GeoMapApp (www.geomapapp.org, accessed on 15 December 2022).

However, it is suggested in the literature that anomalies in electromagnetic signals could be related to seismic events. Observational evidence is reported by analyzing radio, ionospheric, magnetic, or electric signals in diverse frequency bands, detected with a broad type of instruments at distances ranging from some few to thousands of km and linked to earthquakes with a wide range of magnitudes [22–52]. Diverse types of physical mechanisms have been proposed to explain them [53–61]. Some authors explain electromagnetic anomalies based on the electrokinetic effect, in which the solid rock becomes electrically charged. In contrast, the liquid phase in the rock's porosity acquires the opposite charge [53]. Even small changes in the thermal conditions inside the rock mass may promote convection of the liquid phase, generating an electric current that can induce magnetic fields. By observing the thermal contrasts on the surface in the study area (Figure 5), it is necessary to complement the previous explanation to account for the temporal behavior of the electromagnetic anomalies by tying together the stress field, which promotes variable gradients in fluid migration. As the pore pressure increases before intense seismic events, the rocks dehydrate and lose conductivity, which can be measured in significant depths using the MT method. These fractures may cause the breaking of ionic bonds, generating changes in the potential difference and, consequently, electromagnetic signals [53,61].

We hypothesize that these electromagnetic signals respond to changes in the resistivity field of the medium, which in turn is related to changes in the porosity and microfracture conditions and the volume of fluids contained, as suggested in Figure 6. Given changes in the deviatoric stress that generates earthquakes, changes in the porosity field, pore pressure gradients (up to distances greater than the size of the seismic source), and fluid mobility are expected. Thus, our reported changes in $\rho_a$ and the latter observance of gas emissions

on the surface (Figures 3 and 4) constitute possible evidence of the fluid migration in the upper lithosphere during the seismic swarm. Additional confirmation of the phenomena suggested in this work could come from studies that analyze the elastic and inelastic fields using seismic swarm data (e.g., [61–63]). We believe that permanent MT monitoring could be an interesting strategy for understanding the pore pressure conditions before fracture initiation in strong earthquakes or seismic swarm scenarios.

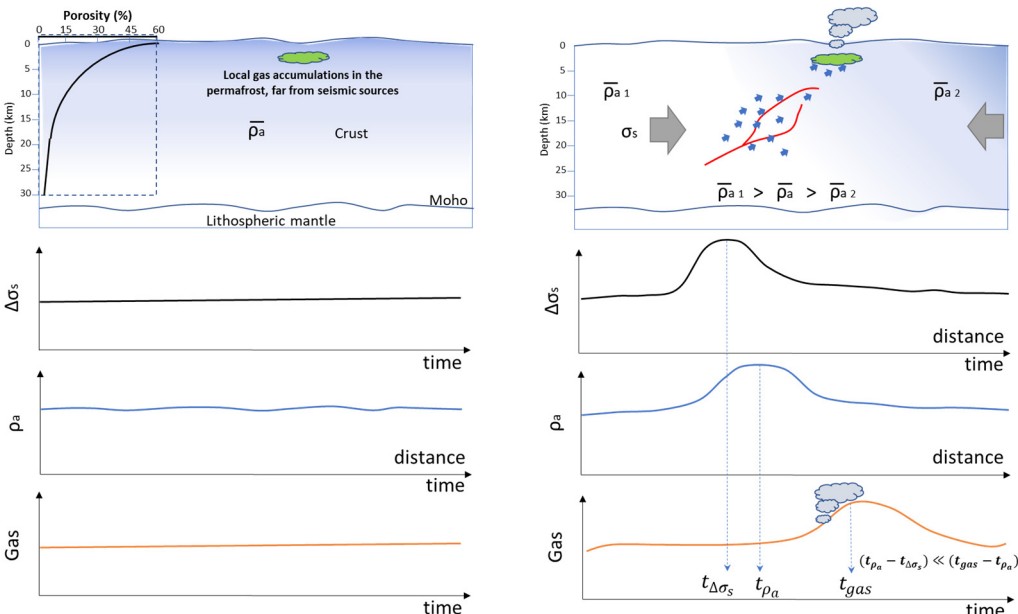

**Figure 6.** Schematic representation of the generation processes previous to (left) and after the seismic swarm. With no significant temporal variations in the tectonic stress ($\Delta\sigma_s$), no changes are expected in $\rho_a$ nor in fluid migration that triggers pore pressure and gas emissions. In contrast, the relevant variations of the $\Delta\sigma_s$ in short times promote variations of porosity, $\rho_a$, pore pressure (suggested by the faded shift of the blue background to the right of the top right panel), fluid migration (blue arrows), and gas emissions from local gas accumulations in the permafrost (represented by a green cloud). Time onsets of these processes could meet the following rule: $\left(t_{\rho_a} - t_{\Delta\sigma_s}\right) \ll \left(t_{gas} - t_{\rho_a}\right)$.

In this work, we report an experiment near the Antarctic Peninsula, a region of very low anthropogenic electromagnetic noise and gas emission contamination, with low solar activity during the recorded dataset, and a relevant scenario generated by an isolated seismic swarm in an area of active tectonics. Even under these particular circumstances and when we are not inverting the 1D–$\rho_a$ profile, uncertainties in the estimation of $\rho_a$ could come from large periods analyzed, which means related to the deeper crustal structure. A future potential solution to this issue could be to stack signals from other near MT stations to consolidate space–time 1D–$\rho_a$ maps that may reinforce the trusty anomalies. In addition, other gas measurements around the study region may clarify the role of the fluid migration paths involved in the dynamics of closing and opening of the porosity field.

Finally, we suggest several possible outlooks that the scientific community may address in the future for this type of research: (1) Deploy arrays of permanent MT stations in areas of high tectonic activity that allow the consolidation of datasets on the relationship between seismicity and crustal $\rho_a$ anomalies. (2) In areas of active magmatism, in addition to deploying MT instruments at different distances, it could be necessary to install monitoring networks for fluid pressure and gas emission sensors to verify the hypothesis of fluid migration and pore pressure that trigger seismicity. (3) Design numerical experiments that allow inferring stress conditions, fluid volumes, and changes in the petrophysical properties of the crust necessary to reproduce $\rho_a$ anomalies, such as those reported in this work.

## 5. Conclusions

One MT station located at Seymour–Marambio Island allowed estimating of the multi–temporal 1D–$\rho_a$ structure in the upper lithosphere of the Antarctic Peninsula. The survey detected $\rho_a$ anomalies that changed over time and were related to a surficial earthquake swarm in the Bransfield Strait, south of King George Island, under the Orca submarine volcano. We detected a shallowing of the $\rho_a$ anomalies that matched with contrasting emission measurements of $CH_4$ and $CO_2$, suggesting that anomalies could be linked with fluid migration and propagation of pore pressure that triggered the release of gases. We hypothesize that before the occurrence of earthquakes, the stress field generates pore pressure gradients from sites close to the seismic source to distances greater than the size of the seismic source, promoting alterations in fluid migration that change the resistivity of the upper lithosphere.

**Author Contributions:** Conceptualization, C.A.V.; methodology, C.A.V. and A.C..; software, J.M.S.; validation, C.A.V., A.C and J.M.S.; formal analysis, C.A.V., A.C., A.M.G., J.V. and J.M.S.; investigation, C.A.V., A.C., A.M.G., J.V. and J.M.S.; resources, C.A.V. and A.C.; data curation, C.A.V., A.C. and J.M.S.; writing—original draft preparation, C.A.V. and A.C.; writing—review and editing, C.A.V., A.C., A.M.G., J.V. and J.M.S.; visualization, C.A.V. and J.M.S.; supervision, C.A.V. and A.C.; project administration, C.A.V. and A.C.; funding acquisition, C.A.V. and A.C. All authors have read and agreed to the published version of the manuscript.

**Funding:** This research received no external funding.

**Institutional Review Board Statement:** Not applicable.

**Informed Consent Statement:** Not applicable.

**Data Availability Statement:** Once the paper is approved, the dataset will be openly available.

**Acknowledgments:** We thank the Universidad Nacional de Colombia for providing laboratories and informatic resources during data acquisition and for preparing this paper. We thank LAMBI (IAA/DNA) personnel. A.M.G. is a member of the Carrera del Investigador Científico CONICET. We thank the Special Issue Editors, Alexey Zavyalov and Eleftheria E. Papadimitriou, as well as two anonymous reviewers, for their valuable comments and suggestions.

**Conflicts of Interest:** The authors declare no conflict of interest.

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
