# Peer review of "Evidencing Fluid Migration of the Crust during the Seismic Swarm by Using 1D Magnetotelluric Monitoring"

_applsci, doi:10.3390/app13042683_

Round 1
Reviewer 1 Report
The authors applied multi-temporal 1D MT surveys to identify space-time anomalies of apparent resistivity in the upper lithospherie beneath the Antarctic Peninsula. An obvious resistivity anomaly was associated with the contrasting earthquake activity, and the physical origin of the anomaly was interpreted as the response to the porosity changes and the consequent fluid migration near the initiation of the rock damage that generates the related earthquake. In the literature, many researches reported that the electromagnetic responses of the earthquakes. Unlike previous ones, this study tried to detect the practical response of the resistivity anomaly to the earthquake. Moreover, a reasonable interpretation for the earthquake-related resistivity anomaly was proposed. The topic illustrated by this paper is very interesting. However, in my personal view, there are some points need to be further improved, which are provided as follows. These comments are just given to the authors for reasonable consideration.
(1) In the 71th line of page 2, the expression “+-60 uT” should be “±μT”.
(2) As for the observing system, the heights of the four electrodes are suggested to be provided. Does the altitude difference meet the observation requirement?
(3) As seen from Fig. 2(b), the two base lines are not orthorhombic and also not directing to the south-north & west-east, so the question is that how much the influence of this observing system on the resistivity results.
(4) In the 95~96 lines of page 4, the first sentence seems incomplete.
(5) As for the Eq. (1), I suggest that the accurate formula is provided first and then the related parameters are set to be suitable values. Otherwise, we don’t know the reasons of the value of 500.
(6) As for the Eq. (2), on the one hand, the “i, j = x, y” should be provided, on another hand, the symbols “i” in Eq. (2) and Eq. (3) conflicting with each other. Thus the symbols “i” in the Eq. (3) should not be expressed by the italic. Besides, the physical meanings of the “μ0” and “ω” should be given.
(7) As for the Kp index, the basis or the evidence of classification is suggested to be added.
(8) As for the Fig. 3(b), why there is no negative value? And how to interpret the zero region (marked by the white) surrounding by the high anomaly signals (marked by the black)? How to interpret the anomaly signals in the deep in Fig. 3(a)?
(9) In Fig. 4, there is a sudden increase for CO2 gas, however, the variation of the CH4 is not clear. Why?
(10) I suggest that the distances and the azimuth angles of the earthquakes are marked in the Fig. 4(b).
(11) As for the heat flux data, the corresponding reference(s) or the data origin should be provided.
(12) According to the physical interpretation model as shown in Fig. 7, the related fault(s) is the key. Thus, I suggest the existing faults to be marked in the Fig. 6. If possible, the Bouguer gravity anomaly map is also suggested to be added for supporting the necessary faults. This is also the reason that the distances and the azimuth angles of the earthquakes are marked in the Fig. 4(b).
(13) Some future works or outlooks are suggested to be supplied at the end of the main text.
(14) The format of the listed references is not unified (e.g., No. 20 and No. 21) and thus the listed references should be checked one by one.
Author Response
The authors applied multi-temporal 1D MT surveys to identify space-time anomalies of apparent resistivity in the upper lithospherie beneath the Antarctic Peninsula. An obvious resistivity anomaly was associated with the contrasting earthquake activity, and the physical origin of the anomaly was interpreted as the response to the porosity changes and the consequent fluid migration near the initiation of the rock damage that generates the related earthquake. In the literature, many researches reported that the electromagnetic responses of the earthquakes. Unlike previous ones, this study tried to detect the practical response of the resistivity anomaly to the earthquake. Moreover, a reasonable interpretation for the earthquake-related resistivity anomaly was proposed. The topic illustrated by this paper is very interesting. However, in my personal view, there are some points need to be further improved, which are provided as follows. These comments are just given to the authors for reasonable consideration.
(1) In the 71th line of page 2, the expression "+-60 uT" should be "±μT".
RESPONSE: Done.
(2) As for the observing system, the heights of the four electrodes are suggested to be provided. Does the altitude difference meet the observation requirement?
RESPONSE: The sentence was extended to clarify this issue. The new sentence starts at line 70th, and it is as follows: "It contains sensors to measure the magnetic and electric fields (Figure 2c), such as a Bartington Mag648L triaxial magnetometer with low noise, a range of +-60T (resolution of 0.012 nT/count), and four copper ground electrodes of 70 cm each, all located approximately at the same level on a plateau 200 m high."
(3) As seen from Fig. 2(b), the two base lines are not orthorhombic and also not directing to the south-north & west-east, so the question is that how much the influence of this observing system on the resistivity results.
RESPONSE: The MT method has been formulated under quasi-static assumptions of electric and magnetic fields, isotropic propagation, and the validity of Maxwell's formulations in an arbitrary coordinate system (Zhdanov, 2009; Chave and Jones, 2012). Field acquisitions try to maintain the convention of N-S and E-W oriented electrodes and magnetic sensors following this reference system. However, due to obstacles inherent to the terrain or the presence of infrastructure, in practice, it is essential to respect the orthogonality between electrical and magnetic data under any coordinate system to guarantee an adequate estimate of the magnetotelluric response tensor. In this work, care was taken to respect this orthogonality. Additionally, as the study focuses on estimating resistivity variations with time using exactly the same field setup for electrical and magnetic measurements, our assessment of the 1D- structure with time allows verifying the detection of transient anomalies to establish possible relationships with tectonic activity. The following sentence was extended to clarify this issue, which starts at line 77th: "The arrangement of the four electrodes makes up the two almost orthogonal dipoles NNE (124m) and EEN (80m), which maintain the same direction as the magnetometer components."
(4) In the 95~96 lines of page 4, the first sentence seems incomplete.
RESPONSE: The sentence was excluded because it was redundant concerning the previous paragraph.
(5) As for the Eq. (1), I suggest that the accurate formula is provided first and then the related parameters are set to be suitable values. Otherwise, we don't know the reasons of the value of 500.
RESPONSE: The theoretical expression that explains the skin depth has been included, and the meaning of the parameters it involves has been moved apart from the formula (lines 116-120).
(6) As for the Eq. (2), on the one hand, the "i, j = x, y" should be provided, on another hand, the symbols "i" in Eq. (2) and Eq. (3) conflicting with each other. Thus the symbols "i" in the Eq. (3) should not be expressed by the italic. Besides, the physical meanings of the "μ0" and "ω" should be given.
RESPONSE: To express the relationship between electric and magnetic fields adequately, eq. (2), the use of the indices i,j has been omitted. Vector symbols were presented with bold letters and parameters not previously mentioned has been clarified. Terms in eq. (3) are now not italics. Same consideration was incorporated in eq. (4).
(7) As for the Kp index, the basis or the evidence of classification is suggested to be added.
RESPONSE: A sentence on line 147 has been included to invite the reader to review the conceptual information of the Kp index. This new sentence avoids long paragraphs with the details of the geomagnetic datasets and empirical equations used to formulate this concept.
(8) As for the Fig. 3(b), why there is no negative value? And how to interpret the zero region (marked by the white) surrounding by the high anomaly signals (marked by the black)? How to interpret the anomaly signals in the deep in Fig. 3(a)?
RESPONSE: We improve the previous version of figures 3 and 4 to show anomalies in terms of positive and negative values with regard to the apparent resistivity average () for each time window analyzed. Figure artifacts were overcome (white areas surrounded by black patches).
(9) In Fig. 4, there is a sudden increase for CO2 gas, however, the variation of the CH4 is not clear. Why?
RESPONSE: On an updated Figure 4, it has been incorporated the following paragraph for explaining this phenomenon (lines 177-183): “Gases detected by the Marambio Station are typically stored in the permafrost near the measurement instrument [13]. Thus, pore pressure perturbances from the seismic source in the Bransfield Basin could promote their trigger emission in Marambio Island. We also observe that there is a sudden increase in CO2 gas. However, the variation of the CH4 has not the same trend. We speculate that contrasting concentrations of these gases near the measurement instrument may explain this behavior.”
(10) I suggest that the distances and the azimuth angles of the earthquakes are marked in the Fig. 4(b).
RESPONSE: Figures 3 and 4 were improved, incorporating distances and azimuth of the earthquakes projected in the space-time sections of .
(11) As for the heat flux data, the corresponding reference(s) or the data origin should be provided.
RESPONSE: The reference is presented in the caption of Figure 5 ([18]).
(12) According to the physical interpretation model as shown in Fig. 6, the related fault(s) is the key. Thus, I suggest the existing faults to be marked in the Fig. 5. If possible, the Bouguer gravity anomaly map is also suggested to be added for supporting the necessary faults. This is also the reason that the distances and the azimuth angles of the earthquakes are marked in the Fig. 4(b).
RESPONSE: Current tectonic cartography in the study zone has not enough resolution for highlighting faults linked to the seismic swarm occurred between August and November, 2020. However, to incorporate suggestions from reviewer, we have modified Figure 6 by including a map with gravity anomalies [16], which highlights the Bransfield Basin. The second map of the figure maintain the original heat flow distribution [18]. Both maps were improved with tectonic features cartographied by [17] with high-resolution swath-bathymetry data. First paragraph of the Dicussion section was improved by incorporating sentences regarding the seismic swarm and its relationship with properties presented in this figure.
(13) Some future works or outlooks are suggested to be supplied at the end of the main text.
RESPONSE: Thank you for this very important suggestion. Ending the Discussion section, we have incorporated several possible outlooks that the scientific community may address in the future for this type of research, such as: 1) Deploy arrays of permanent MT stations in areas of high tectonic activity that allow the consolidation of datasets on the relationship between seismicity and crustal anomalies. 2) In areas of active magmatism, in addition to deploying MT instruments at different distances, it could be necessary to install monitoring networks with fluid pressure and gas emission sensors to verify the hypothesis of fluid migration and pore pressure that trigger seismicity. 3) Design numerical experiments that allow inferring stress conditions, fluid volumes, and changes in the petrophysical properties of the crust necessary to produce anomalies, such as those reported in this work.
(14) The format of the listed references is not unified (e.g., No. 20 and No. 21) and thus the listed references should be checked one by one.
RESPONSE: Done.
Reviewer 2 Report
The manuscript reported that the potential relationship between the conductivity and earthquakes by utilizing the MT data. The observation results show the MT would be a powerful tool to monitor pre-earthquake anomalous phenomena. However, numerous scientific questions have to be resolved before the manuscript can be published in future. The major comments include that references cited in the manuscript are insufficient; the statements and conclusion cannot be entirely supported by the evidence proposed from the author. I would like to suggest the authors carefully revise the manuscript following the comments below.
Major comments
1. References cited in the manuscript are insufficient. Numerous papers related to electromagnetic anomalies before earthquakes. However, these papers do not be cited in the manuscript.
2. The authors have to show that several major factors dominate changes in electromagnetic fields during earthquakes. Traditional works are obtained from the MT. Changes in electrical resistivity monitored by MT could be related to earthquake occurrence.
3. L28 What is the energy of the earthquake?
4. Fig. 1 shows that the MT station is located about 200 km away from earthquakes. Meanwhile, these earthquakes, in general, have small magnitude. This suggests the radius of earthquake preparation zones is generally smaller than 100 km. In short, changes in electrical resistivity are not irrelevant to these earthquakes, but beneath the MT station.
5. I would like to suggest that authors add a new figure to exhibit the relationship between frequency and monitoring depths for the MT station.
6. The seismic swarm is not detected by the authors. Please cite references.
7. What is the magnitude scale of Mww?
8. It is very difficult to find the kp index from Figure 3a. Two different parameters share the same colorbar?
9. Fig. 3b and Fig. 4b are almost the same. Please remove one or combine Figs. 3 and 4 as one.
10. In fact, the concentration of CH4 and CO2 dose not relate to earthquake occurrence (Fig. 4). Therefore, how the authors concluded the model of Fig. 7b.
11. Numerous studies have reported that ground vibrations and strain changes before earthquakes. How do the authors make the assumption of “no variations in the tectonic stress?
12. We agree that micro-cracks develop during earthquakes. However, how can a tunnel from > 10 km in depth.
Author Response
The manuscript reported that the potential relationship between the conductivity and earthquakes by utilizing the MT data. The observation results show the MT would be a powerful tool to monitor pre-earthquake anomalous phenomena. However, numerous scientific questions have to be resolved before the manuscript can be published in future. The major comments include that references cited in the manuscript are insufficient; the statements and conclusion cannot be entirely supported by the evidence proposed from the author. I would like to suggest the authors carefully revise the manuscript following the comments below.
Major comments
- References cited in the manuscript are insufficient. Numerous papers related to electromagnetic anomalies before earthquakes. However, these papers do not be cited in the manuscript.
RESPONSE: We have incorporated many references (> 40) cited in the Discussion section to support observational evidence on earthquake-related electromagnetic phenomena. In this section, it is commented that anomalies from radio, ionosphere, magnetic and electrical signals in various frequency bands, detected with a wide range of instruments at distances from a few to thousands of km from the seismic source, have been linked to earthquakes in a broad range of magnitudes.
- The authors have to show that several major factors dominate changes in electromagnetic fields during earthquakes. Traditional works are obtained from the MT. Changes in electrical resistivity monitored by MT could be related to earthquake occurrence.
RESPONSE: It is correct. We explain the space-time variation of the electrical resistivity of the lithosphere to changes in the field of tectonic stresses that trigger seismic events. We hypothesize that before the occurrence of earthquakes, the stress field generates pore pressure gradients from sites close to the seismic source to distances greater than the size of the seismic source, promoting alterations in fluid migration that change the resistivity of the upper lithosphere.
- L28 What is the energy of the earthquake?
RESPONSE: This term is used to mean the earthquake’s magnitude. In the context of the paragraph, we have excluded using the term magnitude to avoid mentioning of particular scales, e.g., mb, Mw, Mww, MS, mc, md, etc.
- Fig. 1 shows that the MT station is located about 200 km away from earthquakes. Meanwhile, these earthquakes, in general, have small magnitude. This suggests the radius of earthquake preparation zones is generally smaller than 100 km. In short, changes in electrical resistivity are not irrelevant to these earthquakes, but beneath the MT station.
RESPONSE: As is mentioned in the previous answer (2), we hypothesize that before the occurrence of earthquakes, the stress field generates pore pressure gradients from sites close to the seismic source to distances greater than the size of the seismic source (in our case, around 200 km away), promoting alterations in fluid migration that change the resistivity of the upper crust around the Marambio Station. The seismic swarm reported by [8] is linked with the emplacement of a significant volume of magma and fluids that could modify the pore pressure field at large distances. Hence, although Figure 1 only shows events with mb>4.0, the large amount of seismic energy released during the seismic swarm (>85.000 events in a few months) suggests that it is not clear the relevance of small events in the context of changes in electrical resistivity inside a great tectonic-magmatic event as is reported by [8]. A key point for future research is to evaluate the efficiency of the pore pressure gradient on diverse distances via changes in the resistivity in the lithosphere.
- I would like to suggest that authors add a new figure to exhibit the relationship between frequency and monitoring depths for the MT station.
RESPONSE: Figures 3 and 4 present lateral scales associated with the resistivity or their anomalies. One of them (left) presents the period estimated (which is proportional to the depth). Thus, the frequency suggested is implicit in the figure.
- The seismic swarm is not detected by the authors. Please cite references.
RESPONSE: We have changed the reference [8] to present the most updated reference that explains the seismic swarm, which mentions several details of interest that are commented in the paper. The reference is [8] Cesca, S., Sugan, M., Rudzinski, Ł. et al. Massive earthquake swarm driven by magmatic intrusion at the Bransfield Strait, Antarctica. Commun Earth Environ, 2022, 3, 89. https://doi.org/10.1038/s43247-022-00418-5.
- What is the magnitude scale of Mww?
RESPONSE: This magnitude is defined as the Moment W-phase, derived from a centroid moment tensor inversion of the W-phase (~50-2000 s; passband based on the size of EQ). According to Hayes et al. (2009), it is computed for all M5.0 or larger earthquakes worldwide but generally robust for all M5.5 worldwide. Provides consistent results to M~4.5 within a regional network of high-quality broadband stations. We have included its meaning and its seminal reference at line 156.
- It is very difficult to find the kp index from Figure 3a. Two different parameters share the same colorbar?
RESPONSE: The updated version of figures 3 and 4 present a new panel for giving this information. Hence, the new index panel is now easy to visualize and compare with the resistivity and anomalies.
- Fig. 3b and Fig. 4b are almost the same. Please remove one or combine Figs. 3 and 4 as one.
RESPONSE: The updated version of Figure 3 incorporates this suggestion. New Figure 4 becomes a detailed window for explaining the shallowing of the resistivity anomalies linked with the gas emission process.
- In fact, the concentration of CH4and CO2 dose not relate to earthquake occurrence (Fig. 4). Therefore, how the authors concluded the model of Fig. 7b.
RESPONSE: As is mentioned in the previous answer (2), we hypothesize that before the occurrence of earthquakes, the stress field generates pore pressure gradients from sites close to the seismic source to distances greater than the size of the seismic source (in our case, around 200 km away), promoting alterations in fluid migration that change the resistivity of the upper crust. Thus, changes in resistivity and the later observance of gas emissions on the surface constitute possible evidence of a local process triggered by the pore pressure gradients. Those gases are initially stored under or near the measurement instrument. However, as gases are compressible and have different diffusion coefficients, the gas emission does not occur immediately to resistivity anomalies or changes in the local pore pressure gradient. Gases reach the surface slower than deeper anomalies, as suggested in Figure 4. The new Figure 6 suggests a delay times between the onsets of and gas emissions.
- Numerous studies have reported that ground vibrations and strain changes before earthquakes. How do the authors make the assumption of "no variations in the tectonic stress?
RESPONSE: We have revised the sentence to avoid misunderstanding in the context of the paragraph. The new version of the figure caption is as follows: “Schematic representation of the previous (left) and after the earthquake generation process. With no significant temporal variations in the tectonic stress (), it is expected no changes in neither fluid migration that triggers pore pressure and gas emissions. In contrast, the relevant variations of the in short times promote variations of porosity, , pore pressure (suggested by the faded shift of the blue background to the right of the top right panel), fluid migration (blue arrows), and gas emissions from local gas accumulations in the permafrost (represented by a green cloud). Time onsets of these processes could meet the following rule: .”
- We agree that micro-cracks develop during earthquakes. However, how can a tunnel from > 10 km in depth.
RESPONSE: Explanations presented in answers 2, 4 and 10 clarify this request.
Round 2
Reviewer 2 Report
The authors have intensely revised the manuscript. The discussion has been extended. There are two minor comments listed in below. I further encourage the authors added the precipitation data for references. On the other hand, an earthquake cycle is generally defined from a major earthquake to the next major event in a particular area. The authors declare that they monitor changes during a seismic swarm. I would like to suggest that “the seismic swarm” replaces “the earthquake cycle” in title. The manuscript can be published after the minor revision.
1. The unit of earthquake magnitude utilized in a manuscript should be the same. For example, “mb” is shown in Abstract. “M” is utilized in Figures.
2. Lines 27-29. Earthquakes energy is not utilized in the two cited references [1-2].
3. I encourage the authors added the precipitation data for references due to that the concentration of the CH4 and CO2 would be affected by the precipitation.
Author Response
On the other hand, an earthquake cycle is generally defined from a major earthquake to the next major event in a particular area. The authors declare that they monitor changes during a seismic swarm. I would like to suggest that “the seismic swarm” replaces “the earthquake cycle” in title. The manuscript can be published after the minor revision.
Response: We replaced the term “earthquake cycle” in the paper with “seismic swarm”.
1. The unit of earthquake magnitude utilized in a manuscript should be the same. For example, “mb” is shown in Abstract. “M” is utilized in Figures.
Response: We have homogenized the earthquake magnitude scale by using only Mww.
2. Lines 27-29. Earthquakes energy is not utilized in the two cited references [1-2].
Response: We have corrected lines 27-29, changing the term earthquake energy by magnitude.
3. I encourage the authors added the precipitation data for references due to that the concentration of the CH4 and CO2 would be affected by the precipitation.
Response: We improved figures 3 and 4, including precipitation. The caption of figure 3 includes now the following sentence: “The extremely low precipitation at the Marambio Station does not seem to affect gas concentrations. Precipitation data were taken from https://power.larc.nasa.gov/data-access-viewer/.”
